# Diagnostic Value of Menstrual Blood Lipidomics in Endometriosis: A Pilot Study

**DOI:** 10.3390/biom14080899

**Published:** 2024-07-24

**Authors:** Natalia Starodubtseva, Vitaliy Chagovets, Alisa Tokareva, Madina Dumanovskaya, Eugenii Kukaev, Anastasia Novoselova, Vladimir Frankevich, Stanislav V. Pavlovich, Gennady Sukhikh

**Affiliations:** 1National Medical Research Center for Obstetrics Gynecology and Perinatology Named after Academician V.I. Kulakov of the Ministry of Healthcare of Russian Federation, 117997 Moscow, Russia; n_starodubtseva@oparina4.ru (N.S.); a_tokareva@oparina4.ru (A.T.); m_dumanovskaya@oparina4.ru (M.D.); e_kukaev@oparina4.ru (E.K.); a_novoselova@oparina4.ru (A.N.); v_frankevich@oparina4.ru (V.F.); s_pavlovich@oparina4.ru (S.V.P.); g_sukhikh@oparina4.ru (G.S.); 2Moscow Center for Advanced Studies, 123592 Moscow, Russia; 3V.L. Talrose Institute for Energy Problems of Chemical Physics, Russia Academy of Sciences, 119991 Moscow, Russia; 4Laboratory of Translational Medicine, Siberian State Medical University, 634050 Tomsk, Russia; 5Department of Obstetrics, Gynecology, Perinatology and Reproductology, Institute of Professional Education, Federal State Autonomous Educational Institution of Higher Education I.M. Sechenov First Moscow State Medical University of the Ministry of Health of the Russian Federation, 119991 Moscow, Russia

**Keywords:** lipid, endometriosis, dried blood spot, mass spectrometry, diagnostic

## Abstract

Endometriosis is a prevalent chronic inflammatory disease characterized by a considerable delay between initial symptoms and diagnosis through surgery. The pressing need for a timely, non-invasive diagnostic solution underscores the focus of current research efforts. This study examines the diagnostic potential of the menstrual blood lipidome. The lipid profile of 39 samples (23 women with endometriosis and 16 patients in a control group) was acquired using reverse-phase high-performance liquid chromatography–mass spectrometry with LipidMatch processing and identification. Profiles were normalized based on total ion counts. Significant differences in lipids were determined using the Mann–Whitney test. Lipids for the diagnostic model, based on logistic regression, were selected using a combination of variance importance projection filters and Akaike information criteria. Levels of ceramides, sphingomyelins, cardiolipins, triacylglycerols, acyl- and alkenyl-phosphatidylethanolamines, and alkenyl-phosphatidylcholines increased, while acyl- and alkyl-phosphatidylcholines decreased in cases of endometriosis. Plasmenylphosphatidylethanolamine PE P-16:0/18:1 and cardiolipin CL 16:0_18:0_22:5_22:6 serve as marker lipids in the diagnostic model, exhibiting a sensitivity of 81% and specificity of 85%. The diagnostic approach based on dried spots of menstrual blood holds promise as an alternative to traditional non-invasive methods for endometriosis screening.

## 1. Introduction

Endometriosis is a chronic inflammatory disease that affects 10% of women of reproductive age. The progressive nature of endometriosis underscores the necessity of early diagnosis to conserve healthcare resources and improve patients’ quality of life through personalized approaches [1,2]. Timely diagnosis is hindered by the nonspecificity of clinical symptoms, which also characterize other diseases [3]. In some cases, endometriosis progresses without symptoms, leading to delays in diagnosis of more than 10 years [4,5].

While laparoscopy combined with histologic confirmation serves as the gold standard for diagnosing endometriosis, boasting a sensitivity of 94% and specificity of 79%, it is highly invasive and can result in significant delays [5,6]. Ultrasonography and magnetic resonance imaging of pelvic organs offer non-invasive diagnostic alternatives, but ultrasonography is operator-dependent and magnetic resonance imaging (MRI) is costly. The World Endometriosis Society has identified the development of reliable non-invasive diagnostic tools as one of the main directions in gynecology [7].

Endometriosis progression involves alterations in eutopic endometrial tissue, as demonstrated by changes in gene expression and the concentrations of proteins such as A, cysteine-rich angiogenic inducer 61, annexin 1, osteopontin, and aromatase P450 [8]. These molecular alterations in tissue profiles lead to changes in uterine fluid, which can be utilized for biomarker discovery [9]. Additionally, venous blood contains 122 potential biomarkers, including angiogenesis and growth factors, apoptosis markers, cell adhesion molecules, high-throughput markers, hormonal markers, immune system and inflammatory markers, oxidative stress markers, microRNAs, and tumor markers [10].

Menstrual blood serves as a non-invasive source of endometrial cells [11,12], which can be analyzed for potential endometriosis markers and easily collected from women using menstrual cups, specialized pads, or smart tampons [5,12,13]. Menstrual blood contains over 350 unique proteins not found in circulating blood or vaginal fluid [14]. Ji et al., in their study, observed increased menstrual blood levels of C-X-C motif chemokine 5 and interleukin-1 receptor antagonist proteins in cases of endometriosis [15].

Additionally, alterations in the metabolomic profiles of tissue and fluid (blood, follicular fluid) occur in cases of endometriosis [16,17,18,19], with lipids playing a significant role in these profile changes [19]. Mass spectrometry analysis methods, coupled with chromatography or direct analysis (surface-enhanced laser desorption/ionization (SELDI), matrix-assisted laser desorption/ionization (MALDI), shotgun), are effective for metabolic profiling of biological samples and have successfully been used in the search for endometriosis markers [19,20,21]. Dried blood spot (DBS) samples require minimal biological material and have been successfully utilized in diagnostic research [22,23]. The aim of our study is to explore the possibilities of diagnosing endometriosis through the lipid profiles of dried menstrual blood spots.

## 2. Materials and Methods

### 2.1. Study Desing

This study involved samples from 23 patients with histologically verified endometriosis stages I-IV, as per the revised American Fertility Society (rAFS) classification, along with 16 patients comprising the control group. Menstrual blood samples from patients with endometriosis were collected prior to surgery. 

Patients were invited on days 2–3 of their menstrual cycle (during the days of heaviest menstrual bleeding). Menstrual blood samples were collected on a gynecological chair, using a Cusco plastic speculum inserted and then removed, allowing the menstrual blood to flow onto a special form with five equally sized compartments. Application was made only on one side of the form, ensuring that each drop of blood fully saturated the form. Drops could be larger than the compartments but never smaller. Blood drops were air-dried for two hours in the dark at room temperature before being placed in an envelope for transportation. The collected sample was then placed in a plastic bag with a zip-lock and stored in a freezer at −20 °C [24,25]. Documentation was completed beforehand, including the patient’s surname, first name, patronymic, age, endometriosis phenotype, outpatient card number, menstrual cycle day, and date of sample collection.

Inclusion criteria:

For all participants:-Women aged 18–45 with regular menstrual cycles;-Signed informed consent;-Residing in Moscow.

For participants in the endometriosis group:-Endometriosis confirmed by surgical intervention or instrumental visualization methods (ultrasound and/or magnetic resonance imaging of the pelvic organs);-The presence of one or more endometriosis symptoms: dysmenorrhea and/or dyspareunia, and/or chronic pelvic pain, and/or infertility.

For participants in the control group:-Absence of infertility (one or more childbirths);-Regular menstrual cycle;-Absence of severe dysmenorrhea;-Absence of endometriosis confirmed by ultrasound investigation.

Exclusion criteria:-Chronic diseases other than endometriosis (diabetes, hypertension, etc.);-Infectious diseases: human immunodeficiency viruses (HIV) and viral hepatitis (B and C);-Hormonal medication intake within 3 months prior to study inclusion.

The main group consisted of 23 women (age 28 (25;33) years; body mass index (BMI) 19.4 (18.1;22.1) kg/m^2^) with external genital endometriosis (superficial endometriosis, ovarian endometriomas, deep endometriosis), identified by magnetic resonance imaging of the pelvic organs and confirmed histologically after laparoscopy. The control group included 16 women (age 27 (24;33) years; BMI 22.7 (19.7;24.9) kg/m^2^). Groups were comparable in age and BMI (*p* > 0.05).

In the main group, in most cases, there was a combination of various forms of endometriosis: superficial endometriosis with ovarian endometriomas in 26.1% (6/23), ovarian endometriomas and deep endometriosis in 43.5% (10/23), superficial endometriosis with deep endometriosis in 17.4% (4/23), and isolated ovarian endometriomas in 13% of cases (3/23). The leading complaint of the majority of patients from the main group was dysmenorrhea (visual analog scale score was 7.0 (5.0;9.0)), as well as abnormal uterine bleeding (heavy menstrual bleeding) and infertility.

### 2.2. Lipid Extraction

The lipid extract was obtained using a modified Folch method extraction procedure. The dried blood spot was removed and placed in a 2 mL microtube. To this, 480 µL of chloroform/methanol (2:1, *v*/*v*) was added, and the tube was vortexed using ultrasound for 15 min. Subsequently, 150 µL of HPLC-grade water was added, and the mixture was vortexed using ultrasound for another 15 min. The tube was then centrifuged at 15,000 rounds per minute for 10 min. After centrifugation, 150 µL of the organic layer was collected in a 1.5 mL Eppendorf tube, and 250 µL of chloroform/methanol (2:1, *v*/*v*) was added to the microtube again. The mixture was vortexed for 5 min and centrifuged at 15,000 rounds per minute for 10 min. Next, 250 µL of the organic layer was collected and added to the previous 150 µL. The lipid extract was dried under nitrogen flow at 40 °C and then redissolved in 150 µL of isopropanol/acetonitrile (1:1 *v*/*v*). The mixture was vortexed for 5 min and centrifuged at 15,000 rounds per minute for 5 min. Finally, 100 µL of the mixture was collected for chromatography–mass spectrometry analysis.

### 2.3. HPLC-MS Analysis

High-performance liquid chromatography coupled with mass spectrometry (HPLC-MS) was conducted using the Dionex Ultimate 3000 system (Thermo Scientific, Waltham, MA, USA) and the Bruker MaXis Impact instrument (Bruker Daltonics, Bremen, Germany). Reversed-phase chromatography separation was achieved using a Zorbax XDB-C18 column (3.5 µm, 150 mm length, 0.5 mm inner diameter, Agilent, Santa Clara, CA, USA) with mobile phase A consisting of an acetonitrile/water mixture (60/40 *v*/*v*) and mobile phase B consisting of isopropanol/acetonitrile/water (90/8/2 *v*/*v*/*v*). Both mobile phases contained modifiers: 0.1% formic acid and 10 mM ammonium formate. The flow rate was set at 35 µL/min for 22 min with a temperature of 50 °C, and the composition of phase B changed during analysis as follows:0–0.5 min: 15% B;Linear gradient from 10% to 99% B over 15 min;99% B for 4 min;Linear gradient from 99% to 15% B over 15 min;15% B for 2 min.

Electrospray ionization parameters were set as follows: a capillary voltage of 4.1 kV for positive ion mode and 3.0 kV for negative ion mode, a spray gas pressure of 0.7 bar, a drying gas flow rate of 6 L/min, and a drying gas temperature of 200 °C. The mass spectrum range was 100–1700 *m*/*z* with a resolution of 50,000. Tandem mass spectrometry was performed using data-dependent analysis with the following parameters: the three most abundant peaks were selected after a full mass scan, and the corresponding ions were fragmented by collision-induced dissociation using a collision energy of 35 eV and a mass exclusion time of 1 min.

In accordance with guidelines provided by the Metabolomics Quality Assurance & Quality Control Consortium (mQACC, https://www.mqacc.org/ (accessed on 16 May 2024)), blank and quality control samples (QC) were included in batch [26]. The quality control samples were prepared by pooling of 10 µL from each study sample.

### 2.4. Data Processing

Data preprocessing and lipid identification were conducted using Koelmel et al.’s pipeline with MzMine 2.3 and LipidMatch 3.0 [27]. The peak areas of the lipids were normalized by the sum of peak areas (see Appendix A).

Statistically significant differences in lipids were identified using the Mann–Whitney test with a threshold of *p* < 0.05. Lipids were tested for correlation with stages of endometriosis by a Spearman test with significant factor levels of *p* < 0.05.

Feature selection for the diagnostic model was performed based on a combined dataset of negative ion mode and positive ion mode. This was achieved by a combination of variance important projection (VIP) filtering in orthogonal projection to latent structures (OPLS) and subsequent feature selection in logistic regression based on Akaike information criteria (AIC) [28,29]. The dependent value was assigned as 0 for the control group and 1 for the endometriosis group. Relative levels of lipids are presented as Me(Q1;Q3), where Me is median, Q1 is the first quartile, and Q3 is the third quartile.

A threshold value for the dependent variable was calculated based on the results of receiver operating characteristic (ROC) curve analysis from cross-validation control. This analysis was conducted through 100 repetitions of a “train”/”test” split of 70/30. The threshold value was determined as the value that maximized the sum of sensitivity and specificity.

Search of metabolic pathways, associated with endometriosis, was performed by ConsenscusPathDB http://cpdb.molgen.mpg.de/ (accessed on 16 May 2024) [30] with over-representation analysis based on Wikipathways, SMPDB, KEGG, Reactome, PID, Biocarta, Ehmn, Humancyc, INOH, Netpath, and Signallink.

## 3. Results

### 3.1. Dried Menstrual Blood Lipid Profiling

In positive and negative ion modes, 98 and 107 lipids were identified, respectively (Appendix A). These lipids belong to various classes, including (lyso)phosphatidylcholine, (lyso)phosphatidylethanolamine, cardiolipins, plasmanyl- and plasmenyl lipids, oxidized lipids, cholesterol esters, sphingomyelins, and triacylglycerols. In positive ion mode, almost half of the lipids have between-injection variation of less than 10%. In negative ion mode, more than a half of the lipids exhibited between-injection variation of less than 10%. Moreover, the levels of 40 lipids showed statistically significant differences (*p* < 0.05) between endometriosis and control groups.

In the endometriosis group, menstrual blood ceramides, cardiolipins, oxidized lipids, phosphatidylcholine (PC) 16:0_18:0, (plasmenyl-)phosphatidylethanolamines, plasmanyl-phosphatidylcholines, sphingomyelins, and triacylglycerols were increased. Conversely, the levels of (plasmenyl-)phosphatidylcholines, sphingomyelin (SM) d18:0/18:0, and ceramide-1-phosphate (CerP) d18:0/22:0 decreased in the endometriosis cases (Figure 1A and Table 1). Figure 1A presents the volcano plot of lipids identified, emphasizing the potential markers with a fold change between endometriosis and control samples more than 1.5. A number of lipids exhibited statistically significant correlation with disease severity (stage) (Figure 1B).

Over-representation analysis revealed 168 statistically significant (false discovery rate (FDR) < 0.05) enriched metabolic pathways (Appendix A). Glycerophospholipid biosynthesis/metabolism, sphingolipid de novo biosynthesis/metabolism, ceramide signaling, immune system, and acyl chain remodeling of cardiolipin/phosphatidylethanolamine/phosphatidylcholine present the most disturbed processes in endometriosis.

### 3.2. Diagnostic Model Creation

In the OPLS model utilized for feature preselection, the proportion of described dependent values was 63.2%, and the proportion of predicted dependent values was 42.2% (see Appendix A). The logistic regression model was finalized with the inclusion of cardiolipin CL 16:0_18:0_22:5_22:6 and plasmenylethanolamine PE P-16:0/18:1 (Figure 2) in the following equation: (1)$$y=\frac{1}{1+e^{-(5.48+3.35*I_{CL16:0\_18:0\_22:5\_22:6}-2.18*I_{PEP-16:0/18:0})}}$$
where *I*_x_ represents the relative part of the total ion chromatogram (TIC) as a percentage of compound x. 

An optimal threshold value of 0.59 was determined, resulting in a sensitivity of 81% and a specificity of 85% (see Figure 3).

The level of cardiolipin CL 16:0_18:0_22:5_22:6 was markedly increased in menstrual blood of patients with endometriosis (FC (fold change) = 3.3, *p* = 0.001). Its significance in the equation corresponds to the calculated fold change in the median. Therefore, the menstrual blood from patients with endometriosis exhibits a higher level of cardiolipin CL 16:0_18:0_22:5_22:6 and a lower level of plasmenylphosphatidylethanolamine PE P-16:0/18:1 (Figure 4). Moreover, cardiolipin CL 16:0_18:0_22:5_22:6 was positively correlated with endometriosis stage (R = 0.56, *p* < 0.001). HPLC-MS analysis reliability may be confirmed by low QC deviation: for cardiolipin CL 16:0_18:0_22:5_22:6, this parameter equals 11% (3%) in case of non-normalized (normalized) values, and that for plasmenyl-phosphatidylethanolamine PE P-16:0/18:1 equals 8% (3%), accordingly. 

## 4. Discussion

Endometriosis is not just a medical condition but also a socially significant disease that impacts various aspects of women’s lives. The profound impact of endometriosis goes beyond its physical symptoms and clinical manifestations, affecting women’s emotional well-being, quality of life, relationships, work productivity, and overall societal participation [31]. The chronic nature of the disease, affecting only women of reproductive age, coupled with its unpredictable symptomatology (chronic pelvic pain, dysmenorrhea, dyspareunia, and gastrointestinal symptoms) and potential fertility implications, poses significant challenges in both diagnosis and management due to its chronic inflammatory nature [32]. As a progressive disease, early detection is essential to effectively preserve healthcare resources, enhance patient outcomes, and tailor treatment strategies through personalized approaches. To address the diagnostic challenges posed by the nonspecific symptoms of endometriosis, a multidimensional approach integrating medical history, physical examinations, imaging studies, and minimally invasive procedures such as laparoscopy is crucial for accurate diagnosis [33]. Additionally, the development and validation of specific biomarkers and imaging modalities tailored to identify endometriosis early on could revolutionize diagnostic paradigms and enhance timely intervention.

Venous blood proteins such as vascular endothelial growth factor (VEGF), urocortin, C-reactive protein (CRP), tumor necrosis factor alpha (TNF-alpha), interleukin-6 (IL-6), and follistatin or protein sets can possess particular diagnostic strength [10], but their abnormal levels may also be caused by other diseases [34]. miRNA biomarkers also exhibit high sensitivity and specificity [35] but they entail high-cost analysis. Lipidomics, the comprehensive study of lipid molecules in biological systems, has emerged as a valuable tool in understanding the intricate interplay between lipids and various diseases [36]. Lipids play crucial roles in various cellular processes, including inflammation, immune response, and hormonal regulation, all of which are implicated in the pathogenesis of endometriosis [37]. By applying lipidomics approaches, researchers can dissect the lipid profiles in endometrial tissue, peritoneal fluid, blood, and other biological samples from individuals with endometriosis, offering deep insights into the lipid alterations associated with the disease. Studies employing lipidomics have highlighted dysregulated lipid metabolism in endometriosis, including changes in phospholipids, glycerolipids, sphingolipids, and sterols [38,39,40,41,42,43]. Furthermore, lipidomics investigations have unveiled the role of bioactive lipids, such as prostaglandins and leukotrienes, in modulating inflammatory responses and pain perception in endometriosis [42,44]. Targeting specific lipid pathways through pharmacological interventions or dietary modifications may offer novel therapeutic strategies for managing endometriosis-related symptoms and improving patient outcomes [5]. Overall, the integration of lipidomics in endometriosis research provides a comprehensive understanding of lipid dysregulation in the disease pathogenesis, paving the way for personalized medicine approaches, biomarker discovery, and the development of targeted therapies tailored to individual lipid profiles.

Menstrual blood, known for its ease of collection and non-invasive nature, represents an abundant reservoir of endometrial cells [12]. The unique composition of menstrual blood, enriched with shed endometrial tissue during the menstrual cycle, presents a rich source of cells that reflect the dynamic changes occurring in the endometrium [45,46]. This biological fluid harbors a diverse array of cell types, including epithelial cells, mesenchymal stem cells, endothelial cells, and immune cells, each with distinct functional properties and regenerative potential [47]. Moreover, the non-invasive nature of collecting menstrual blood samples makes it an attractive option for longitudinal studies, biomarker discovery, and monitoring disease progression over time [11,12]. It is noteworthy that menstrual blood is the least researched sample in endometriosis research [35]. The convenience of collecting menstrual blood through self-collection using a menstrual cup and subsequently transferring it onto filter paper for air drying presents dried menstrual blood spot samples as an excellent screening material [5,12,13]. 

Dried blood spots (DBSa) are a sampling method used in screening programs to collect small, defined volumes of blood. This technique offers several advantages for screening purposes. The use of DBS enables easy collection, storage, and transportation of blood samples, as the dried spots on filter paper are stable at room temperature for extended periods [48]. This characteristic simplifies sample handling logistics and allows for decentralized sample collection, making it feasible to conduct screening programs in remote areas or community settings without sophisticated laboratory infrastructure [49]. Furthermore, DBS sampling can facilitate high-throughput screening efforts due to the ease of processing multiple samples in a cost-effective manner. Automation and standardized protocols for punching, eluting, and analyzing DBS samples can streamline the screening workflow, increase efficiency, and reduce manual labor requirements compared to traditional liquid blood sample processing methods [50]. Moreover, DBS samples are suitable for a wide range of screening tests, including genetic screening, infectious disease detection, drug monitoring, and biomarker analysis [51]. The versatility of DBS makes it a valuable tool in population-based screening programs aimed at early detection, disease prevention, and monitoring of at-risk individuals.

This study is a pioneering effort in conducting lipid profiling of menstrual blood. While previous research has primarily focused on the protein profile of menstrual blood in relation to endometriosis, our findings reveal substantial alterations in the lipidome of menstrual blood among patients with endometriosis, encompassing fifteen lipid classes. These changes likely signify a disruption in membrane lipids of shedding epithelium and immune cells, along with disturbances in intracellular lipid metabolism.

The observed trends in phosphatidylethanolamines and sphingomyelins align with alterations seen in the ectopic endometrium compared to the eutopic endometrium [52]. Notably, the levels of unsaturated phosphatidylcholines decrease in menstrual blood, in contrast to their elevation in the ectopic endometrium relative to the eutopic endometrium. Furthermore, the eutopic endometrium in patients with endometriosis also demonstrates pathological changes, underscoring the complex interplay between lipid profiles and the pathophysiology of this condition [52]. Phosphatidylcholines PC 16:0_18:1, PC 18:1_18:2, and PC 18:0_18:2 are decreased in menstrual blood samples in cases of endometriosis, consistent with phosphatidylcholines with the same length and saturation degree showing a similar direction of alteration in peritoneal fluid [43]. 

The level of plasmalogens was dramatically disturbed in the endometriosis group. Moreover, plasmenyl-phosphatidylethanolamine PE P-16:0/18:1 was included in the diagnostic model. Research on the role of plasmalogens in this pathology is still in its early stages. However, Vouk et al. included plasma and peritoneal fluid plasmanylphosphatidylcholines in an endometriosis diagnostic model [53,54]. Studies in other disease contexts have suggested that plasmalogens can modulate inflammatory responses, regulate immune cell function, and protect cells from oxidative damage [55]. Given that chronic inflammation and immune dysregulation are key features of endometriosis, investigating the relationship between plasmalogens and these processes could provide valuable insights into the development and progression of the disease [2,31]. Moreover, plasmalogens possess both anti-inflammatory and antioxidant properties, making them potentially beneficial for alleviating the inflammatory responses and oxidative stress commonly linked to endometriosis [56,57].

Phosphatidylglycerol PG 18:1_18:1 is one marker that increases in menstrual blood in cases of endometriosis, consistent with Feider et al.’s observation of increased levels of this lipid in endometriosis lesions compared to eutopic endometrium [58]. Moreover, we observed a substantial increase in unsaturated triglycerides, mainly with oleic C18:1 and linoleic C18:2 fatty acids. Elevated levels of triglycerides have been associated with obesity, insulin resistance, and metabolic syndrome, all of which are risk factors for endometriosis. Studies have shown a potential link between high levels of triglycerides and an increased risk of developing endometriosis or experiencing more severe symptoms [59,60]. Adipose tissue is an active endocrine organ that secretes various adipokines and inflammatory mediators, which can promote inflammation and affect hormone levels, potentially contributing to the development and progression of endometriosis [61]. Moreover, insulin resistance, which is associated with high triglyceride levels, may also play a role in the pathogenesis of endometriosis [62,63]. Insulin resistance can lead to increased levels of insulin and insulin-like growth factor 1 (IGF-1), which have been implicated in promoting the growth and proliferation of endometrial cells outside the uterus [64,65]. On the other hand, alterations in lipid metabolism, including changes in triglyceride levels, may be consequences rather than causes of endometriosis. Endometriosis itself can lead to systemic inflammation, oxidative stress, and metabolic dysregulation, which could influence lipid metabolism and result in changes in triglyceride levels.

According to our data, endometriosis leads to a significant increase in various fatty acids (such as C16:0, C18:0, C18:1, C18:2, C18:3, C20:3, C20:4) present in lipids like triglyceride, phosphatidylcholine, phosphatidylethanolamine, and cardiolipin. Among these fatty acids, oleic acid C18:1 stands out as particularly prominent. This prevalent fatty acid, commonly found in adipocytes, likely plays a crucial role in fueling the energy needs of the pathologic endometrial cells as they proliferate and metastasize within the body. Interestingly, this lipid profile mirrors a similar pattern observed in other aggressive gynecological neoplasms, including cancer [66]. In particular, accumulation of lipid drops, mainly composed of triglycerides, in tumor-associated macrophages has been observed in a variety of cancers and is strongly associated with poor prognosis [67]. Other fatty acids with significant increase in the endometriosis group are strongly linked with inflammation pathways [68]. Step-by-step convertion of linoleic acid (LA) C18:2 to gamma-linolenic (GLA) C18:3, then dihomo-γ-linolenic acid (DGLA) C20:3 and finally arachidonic (AA) C20:4 acid represents the vital fatty acid metabolism, resulting in a wide range of pro- and anti-inflammatory eicosanoids, such as prostaglandins and leukotrienes [69,70]. Triglycerides rich in arachidonate are actively taken up and stored in lipid droplets within a variety of cell types such as leukocytes, epithelial cells, and neoplastic cells. In a pro-inflammatory enviroment, the lipid droplets may combine all the necessary enzymatic machinery responsible for producing eicosanoids derived from arachidonic acid [70,71]. Increased levels of prostaglandin E2 (PGE2) have been observed in ectopic endometrial tissue, peritoneal fluid, and serum of women with endometriosis, contributing to inflammation and pain associated with the disease [72,73]. Arachidonic acid-derived leukotrienes play a pivotal role in the recruitment and activation of immune cells [74], leading to chronic inflammation and tissue damage in endometriosis. Additionally, the dysregulation of enzymes involved in arachidonic acid metabolism, such as cyclooxygenase-2 (COX-2) and 5-lipoxygenase (5-LOX), has been reported in endometriotic lesions, further supporting the role of arachidonic acid-derived inflammatory mediators in the pathophysiology of endometriosis [75]. Oppositely, eicosanoids derived from DGLA, such as prostaglandin E1 (PGE1) and 15-hydroxyeicosatrienoic acid (15-HETrE), have been shown to exhibit anti-inflammatory properties [76]. These DGLA-derived eicosanoids can counteract the pro-inflammatory effects of arachidonic acid-derived eicosanoids in the body [77]. The balance between arachidonic acid-derived eicosanoids and DGLA-derived eicosanoids is crucial for maintaining optimal inflammatory responses in the body [78,79]. Disruption of this balance, such as in conditions like endometriosis where arachidonic acid metabolism is dysregulated, can lead to chronic inflammation and disease progression.

In this study, cardiolipins were significantly elevated in the menstrual blood of patients with endometriosis. Moreover, very-long-chain cardiolipin CL 16:0_18:0_22:5_22:6 (FC = 3.3, *p* = 0.001) was included in the final logistic regression model for endometriosis diagnosis. Cardiolipins are a unique class of phospholipids found predominantly in the inner mitochondrial membrane, where they play essential roles in mitochondrial structure, function, and cellular metabolism [80]. While the focus on cardiolipins in endometriosis research is not as extensive as with other lipid classes like ceramides, there is growing interest in their potential implications in the pathogenesis of endometriosis [81]. Mitochondrial dysfunction has been implicated in endometriosis, and alterations in cardiolipin content or metabolism could contribute to this dysfunction [82,83,84]. Cardiolipins are crucial for maintaining mitochondrial membrane integrity, regulating energy production through oxidative phosphorylation, and influencing apoptotic pathways [85]. We can only guess why the detected pronounced changes in cardiolipin levels affect minor lipids rather than major molecular species like cardiolipin CL 18:2_18:2_18:2_18:2. Perhaps the reason lies in the disruption in endometriosis of the fine mechanisms regulating the composition of the inner mitochondrial membrane of epithelial cells, shedding during menstruation. Dysregulation of cardiolipin levels or composition may lead to impaired mitochondrial function, increased oxidative stress, and altered apoptotic signaling, all of which are features associated with endometriosis. Moreover, cardiolipins can interact with immune cells and modulate inflammatory responses through the production of cytokines and reactive oxygen species. 

Ceramides and sphingomyelins are identified as lipid species whose levels increase in menstrual blood in cases of endometriosis. The direction of alteration of ceramides Cer d18:0/24:0, Cer d18:1/16:0, Cer d18:1/24:1, and sphingomyelin SM d18:1/24:0 in our study aligns with the alteration of these lipids in peritoneal fluid as observed in Lee et al.’s study, although in endometrial tissue, these lipids exhibit the opposite direction of alteration [86]. Another study has also reported an association between higher concentrations of ceramides in peritoneal fluid and endometriosis progression [87]. Nevertheless, Dominguez et al. found decreased levels of ceramides in endometrial fluid in cases of ovarian endometriosis [41]. Ceramides, a class of sphingolipids, have been implicated in various physiological processes, including cell growth, differentiation, apoptosis, inflammation, and pain modulation [88,89]. In the context of endometriosis, ceramides have gained attention due to their potential role in the modulation of pain sensation [90,91]. Dysregulated ceramide metabolism has been linked to increased inflammation, oxidative stress, and aberrant cell signaling pathways, all of which are key features of endometriosis [87,92,93]. In particular, ceramides have been shown to promote the production of pro-inflammatory mediators and cytokines, exacerbating the inflammatory response in endometriotic lesions [94,95,96,97]. Moreover, ceramides have been implicated in the modulation of pain perception by the activation of nociceptors, the sensitization of peripheral nerves, and the modulation of central pain processing [98,99]. By influencing neuronal excitability and neurotransmitter release, ceramides may contribute to the chronic pain experienced by individuals with endometriosis [100,101,102]. Furthermore, an inflammatory environment can trigger the growth and stimulation of nerve fibers, resulting in the activation of pain signaling pathways [103]. 

The balance between sphingomyelins and ceramides, which are key components of cell membranes and signaling molecules, is critical for cellular function and homeostasis [104]. Sphingomyelins are synthesized from phosphatidylcholines and ceramides through the action of sphingomyelin synthase, primarily in the plasma and Golgi membranes of cells. On the other hand, sphingomyelinases catalyze the breakdown of sphingomyelins back into ceramides [105]. In endometriosis, the dysregulation of sphingolipid metabolism, including alterations in sphingomyelin synthesis and breakdown, may contribute to the observed increase in both ceramides and sphingomyelins. 

Oxidized lipids (in particular, unsaturated phosphatidylcholines and cardiolipins) are also potential markers of endometriosis. Increased lipid oxidation is a hallmark of endometriosis, related to oxidative stress and chronic inflammation [106,107,108,109,110]. In endometriosis, there is a distinct pattern of iron-dependent lipid peroxide buildup and increased resistance to ferroptosis, facilitated by the upregulation of redox enzymes like glutathione peroxidase 4 (GPx4) and superoxide dismutase (SOD) [111]. The presence of oxidized lipids in the peritoneal fluid and endometrial tissues has been reported in pathology [112,113,114,115,116,117,118,119,120]. These lipids can initiate inflammatory responses, activate immune cells, and promote the proliferation and survival of endometrial lesions [106]. Oxidized lipids can also interact with receptors on immune cells, such as Toll-like receptors, leading to the production of pro-inflammatory cytokines and chemokines [121]. Additionally, oxidized lipids have been shown to disrupt cellular signaling pathways, promote oxidative stress, and impair the function of various cellular components, such as mitochondria and membranes [118,122]. These effects can impact cell survival, proliferation, and migration, which are critical processes in the pathogenesis of endometriosis. However, increased levels of oxidized phospholipids may be an artifact of sample collection, namely the long stand after drying. In this work, the drying process was strictly controlled, which allows for minimizing this potential confounder.

## 5. Conclusions

This study unveiled notable variances in the lipid profile of dried menstrual blood spots, indicating its viability as a diagnostic instrument for endometriosis. The lipid biomarker-based model displayed encouraging levels of sensitivity and specificity for diagnosing endometriosis. Delving into the molecular composition of menstrual blood presents exciting possibilities for propelling research on pelvic diseases forward. Continued exploration into the biomarkers present in menstrual blood could potentially usher in enhanced diagnostic approaches and personalized treatment tactics for conditions such as endometriosis.

## Figures and Tables

**Figure 1 biomolecules-14-00899-f001:**
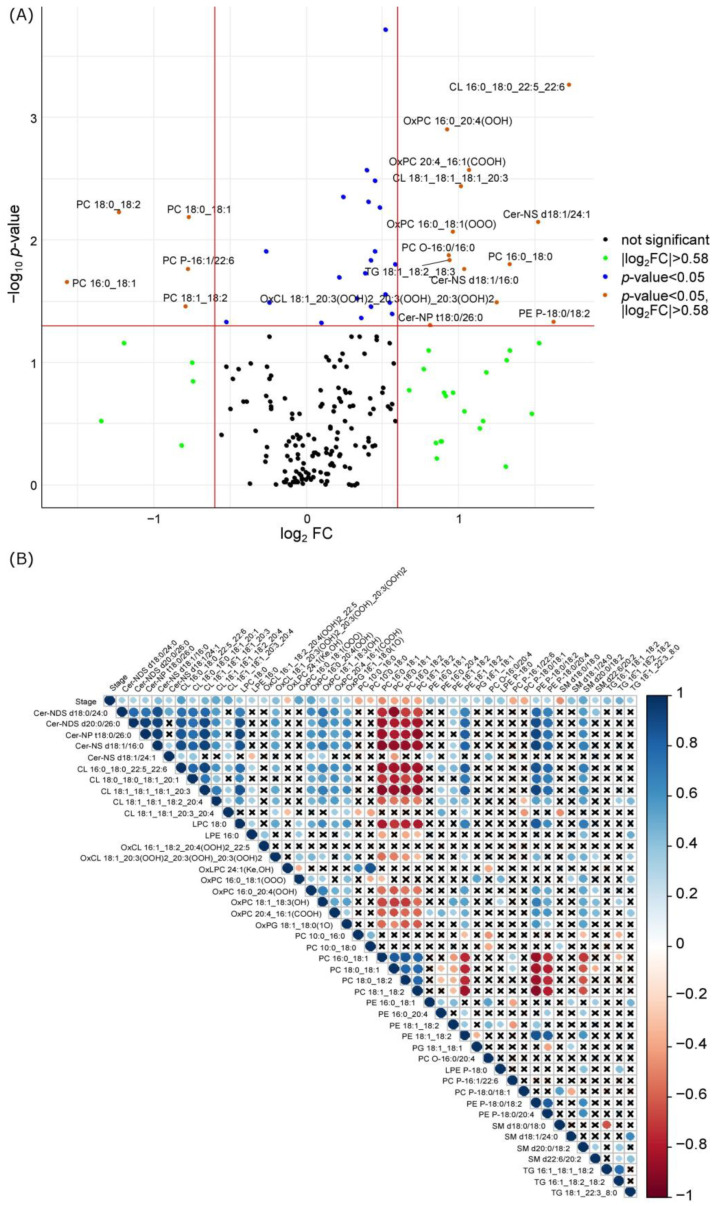
The association of the dried menstrual blood spot lipids with endometriosis. (**A**) A volcano plot of lipids detected in the dried menstrual blood spots. Orange color marks lipids with statistically significant differences between groups studied (*p*-value < 0.05, Mann–Whitney test) and a median fold change greater than 1.5 is labeled. (**B**) Correlation diagram of the lipids with the severity of endometriosis (stage) (*p* < 0.05). X represents the absence of a significant association. Cer-NS—ceramide, Cer-NP—physphingosine ceramide, CL—cardiolipine, oxCL—oxidized cardiolipins, PC—phosphatidylcholine, PC O—plasmanylphosphatidylcholines, PC P—plasmenylphosphatidylcholines, oxPC—oxidized phosphatidylcholines, PE P—plasmenylphosphatidylethanolamine, LPE P —plasmenyllysophosphatidylethanolamine, PG — phosphatidylglycerol, TG —triacylglycerol.

**Figure 2 biomolecules-14-00899-f002:**
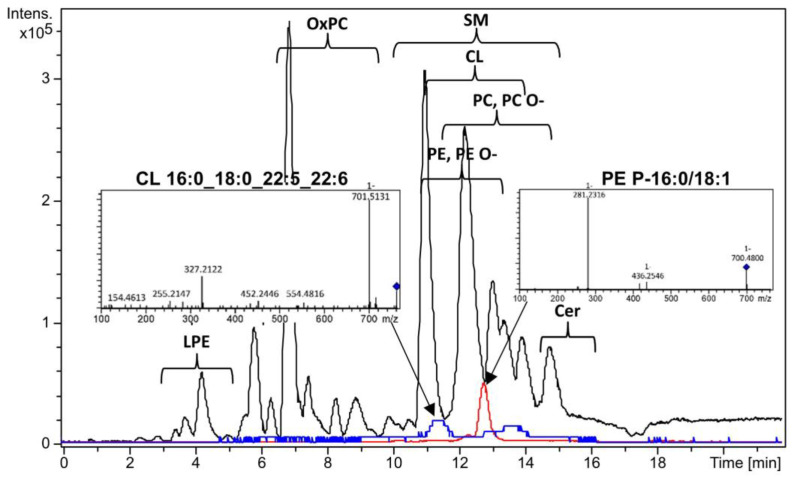
Base peak chromatogram (black), extracted ion chromatogram of PE P-16:0/18:1 (red) and extracted ion chromatogram of CL 16:0_18:0_22:5_22:6 (blue) with tandem mass spectra. Extracted ion chromatogram of CL 16:0_18:0_22:5_22:6 is scaled five times. Negative ion mode.

**Figure 3 biomolecules-14-00899-f003:**
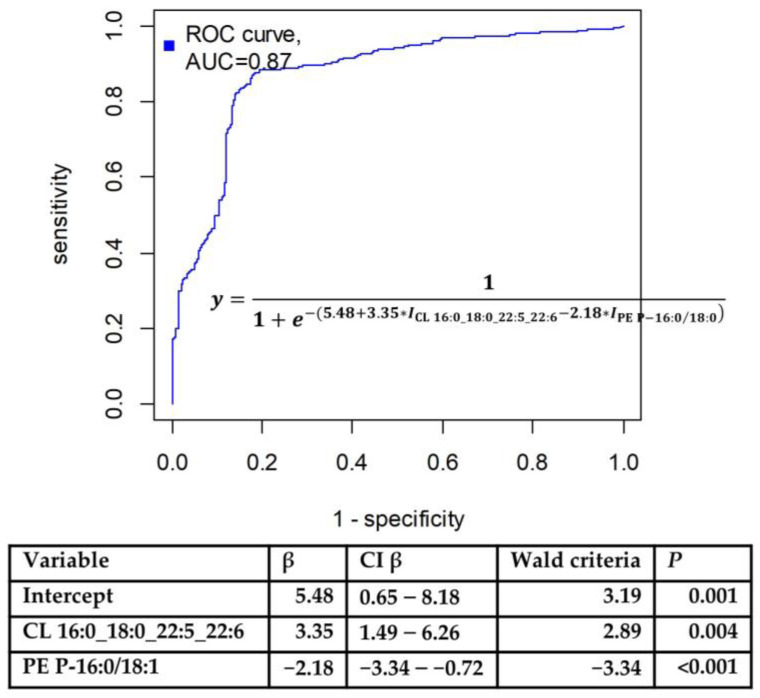
Receiver operating characteristic curve obtained after cross-validation for logistic regression model developed (1). Coefficients in endometriosis diagnostic model, based on logic regression, are presented underneath.

**Figure 4 biomolecules-14-00899-f004:**
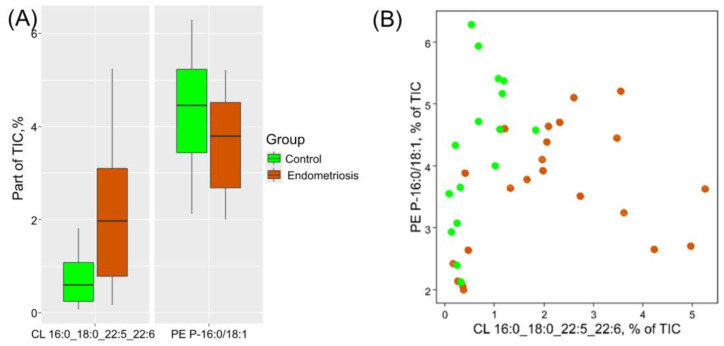
Diagnostic potential of plasmenylphosphatidylethanolamine PE P-16:0/18:1 and cardiolipin CL 16:0_18:0_22:5_22:6: (**A**) Boxplots illustrating the levels of these markers in menstrual blood; (**B**) dried menstrual blood spots plotted in the space of the relative proportion of plasmenyl-phosphatidylethanolamine PE P-16:0/18:1 and cardiolipin CL 16:0_18:0_22:5_22:6. The green points represent samples from the control group patients, while the ocher points represent samples from the endometriosis group.

**Table 1 biomolecules-14-00899-t001:** Relative level of statistically significant different lipids in dried menstrual blood spots, *p*-value (Mann–Whitney test), and fold change in median values. Values are presented as Me(Q1;Q3), where Me is median, Q1 is the first quartile, and Q3 is the third quartile. Cer-NS(NP, NDS)-(phytosphingosine, dihydrosphingosine) ceramide, CerP—ceramide-1-phosphate, CL—cardiolipine, oxCL—oxidized cardiolipins, PC—phosphatidylcholine, PC O—plasmanylphosphatidylcholines, PC P—plasmenylphosphatidylcholines, oxPC—oxidized phosphatidylcholine, PE—phoshpatidyl-ethanolamine, PE P—plasmenylphosphatidylethanolamine, SM—sphingomyelin, TG—triacylglycerol, PG—phosphatidylglycerol.

Lipid	Control	Endometriosis	*p*	Fold Change
Cer-NDS d18:0/24:0	0.58 (0.31;0.68)	0.78 (0.53;1.26)	0.03	1.34
Cer-NP t18:0/26:0	0.68 (0.28;0.91)	1.19 (0.85;1.83)	0.049	1.76
Cer-NS d18:1/16:0	1.30 (0.74;2.04)	2.66 (1.29;3.69)	0.02	2.05
Cer-NS d18:1/24:1	0.11 (0.10;0.36)	0.30 (0.14;0.52)	0.007	2.87
CerP d18:0/22:0	5.08 (3.97;5.70)	3.52 (2.60;5.22)	0.046	0.69
CL 16:0_18:0_22:5_22:6	0.60 (0.24;1.08)	1.97 (0.79;3.09)	0.001	3.31
CL 18:0_18:0_18:1_20:1	0.23 (0.14;0.28)	0.32(0.19;0.47)	0.03	1.43
CL 18:1_18:1_18:1_20:3	0.81 (0.43;1.11)	1.64 (0.94;2.39)	0.004	2.02
CL 18:1_18:1_18:2_20:4	0.25 (0.23;0.32)	0.36 (0.32;0.46)	<0.001	1.44
CL 18:1_18:1_20:3_20:4	0.06 (0.04;0.08)	0.08 (0.07;0.13)	0.005	1.33
OxCL 18:1_20:3(OOH)2_20:3(OOH)_20:3(OOH)2	0.04 (0.03;0.07)	0.10 (0.05;0.18)	0.03	2.38
OxPC 16:0_18:1(OOO)	0.13 (0.09;0.16)	0.24 (0.15;0.37)	0.009	1.95
OxPC 16:0_20:4(OOH)	0.13 (0.07;0.17)	0.24 (0.15;0.30)	0.001	1.90
OxPC 18:1_18:3(OH)	0.18 (0.12;0.28)	0.27 (0.21;0.53)	0.02	1.50
OxPC 20:4_16:1(COOH)	0.05 (0.04;0.08)	0.12 (0.07;0.14)	0.003	2.10
PC 16:0_18:0	1.13 (0.78;2.35)	2.86 (1.25;0.34)	0.02	2.52
PC 16:0_18:1	10.10 (4.54;14.82)	3.41 (1.70;7.90)	0.02	0.34
PC 18:0_18:1	3.38 (2.50;3.90)	1.98 (1.27;3.33)	0.007	0.59
PC 18:0_18:2	5.37 (3.68;6.82)	2.29 (1.38;4.71)	0.006	0.43
PC 18:1_18:2	1.35 (1.06;2.48)	0.78 (0.67;1.30)	0.03	0.58
PE 16:0_18:1	0.32 (0.28;0.35)	0.38 (0.34;0.41)	0.004	1.18
PE 16:0_18:2	0.15 (0.10;0.29)	0.22 (0.16;0.29)	0.04	1.48
PE 16:0_20:4	0.37 (0.30;0.45)	0.51 (0.44;0.57)	0.005	1.40
PE 18:1_18:2	0.12 (0.08;0.16)	0.15 (0.14;0.17)	0.01	1.34
PG 18:1_18:1	0.33 (0.22;0.40)	0.42 (0.31;0.58)	0.04	1.29
PC O-16:0/16:0	0.42 (0.14;0.61)	0.81 (0.45;0.10)	0.01	1.91
PC O-16:0/20:4	0.16 (0.11;0.18)	0.22 (0.19;0.25)	0.003	1.37
PC P-16:1/22:6	0.52 (0.34;0.60)	0.30 (0.24;0.37)	0.02	0.58
PC P-18:0/18:1	0.46 (0.39;0.55)	0.39 (0.34;0.43)	0.03	0.84
PE P-18:0/18:2	0.14 (0.08;0.37)	0.44 (0.19;0.62)	0.046	3.08
PE P-18:0/20:4	2.48 (1.80;2.96)	3.32 (2.56;3.62)	0.03	1.34
SM d18:0/18:0	0.24 (0.17;0.30)	0.20 (0.14;0.23)	0.01	0.83
SM d18:1/24:0	4.05 (3.44;4.76)	5.29 (3.98;6.08)	0.02	1.31
SM d20:0/18:2	0.27 (0.18;0.36)	0.36(0.31;0.61)	0.003	1.32
SM d22:6/20:2	0.09 (0.04;0.12)	0.12 (0.08;0.15)	0.047	1.37
TG 16:1_18:1_18:2	1.72 (1.48;2.03)	2.51 (1.66;2.91)	0.03	1.46
TG 16:1_18:2_18:2	0.28 (0.19;0.33)	0.38 (0.29;0.52)	0.01	1.37
TG 18:1_18:2_18:2	1.48 (0.93;1.78)	1.86 (1.36;2.73)	0.03	1.26
TG 18:1_18:2_18:3	0.58 (0.39;1.21)	1.10 (0.07;1.61)	0.01	1.92
TG 18:1_22:3_8:0	2.18 (1.81;2.36)	2.53 (2.20;2.95)	0.02	1.16

## Data Availability

Data are contained within the Appendix A.

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
