# Peer review of "Diagnostic Value of Menstrual Blood Lipidomics in Endometriosis: A Pilot Study"

_biomolecules, 2024, doi:10.3390/biom14080899_

Round 1

Reviewer 1 Report

Comments and Suggestions for Authors

Authors are presenting interesting study about endometriosis, but I found severe analytical errors in the article. Can authors provide answer to the comments below:

I don't understand why authors use Ultimate Nano LC system and UHPLC Zorbax column? Also, authors do not state the flow rate (Lines 135-142) 

Authors do not mention any quality control samples or blanks, were they used? If not, this would be a severe error in the design of the experiment and the data cannot be trusted, due to unknown analytical errors. The QC samples must be used to correct the analytical variance. 

I would recommend to the authors to follow guidelines provided by mQACC (https://www.mqacc.org/)

Without confirmation and proof of quality control data, the article cannot be published due to the wrong design of the experiment.

Author Response

1.1 I don't understand why authors use Ultimate Nano LC system and UHPLC Zorbax column? Also, authors do not state the flow rate (Lines 135-142)

Answer: Thank you for this remark. It was misprint. It was corrected and the flow rate was added: “High-performance liquid chromatography coupled with mass spectrometry (HPLC-MS) was conducted using the Dionex Ultimate 3000 system (Thermo Scientific, Waltham, MA, USA) and the Bruker MaXis Impact instrument (Bruker Daltonics, Bremen, Germany). Reversed-phase chromatography separation was achieved using a Zorbax XDB-C18 column (3.5 µm, 150 mm length, 0.5 mm inner diameter, Agilent, USA) with mobile phase A consisting of an acetonitrile/water mixture (60/40 v/v) and mobile phase B consisting of isopropanol/acetonitrile/water (90/8/2 v/v/v). Both mobile phases contained modifiers: 0.1% formic acid and 10 mM ammonium formate. The flow rate was set at 35 µL/min for 22 minutes with a temperature of 50°C”.

1.2 Authors do not mention any quality control samples or blanks, were they used? If not, this would be a severe error in the design of the experiment and the data cannot be trusted, due to unknown analytical errors. The QC samples must be used to correct the analytical variance. I would recommend to the authors to follow guidelines provided by mQACC (https://www.mqacc.org/). Without confirmation and proof of quality control data, the article cannot be published due to the wrong design of the experiment.

Answer: Thank you for your kind recommendation to follow the guidelines provided by mQACC (https://www.mqacc.org/). Actually, we used quality control samples and blanks during analysis and added the appropriate information about measures of quality control samples in Supplementary 1 and information about preparation of the quality control samples in the article: “In accordance to guidelines provided by mQACC (https://www.mqacc.org/) blank and quality control samples were included in batch. The quality control samples were prepared by pooling of 10 µl from each study sample.” and “In positive ion mode, almost half of lipids have between injection variation less than 10%. In negative ion mode, more than a half of lipids exhibited between injection variation less than 10%.”.

Reviewer 2 Report

Comments and Suggestions for Authors

The authors have found several lipid molecular species that may define endometriosis by lipidomics analysis of menstrual blood samples. These findings would be novel and important, becasuse there have been many reports on lipidomics using plasma/serum from patients with endometriosis, but not menstrual blood. However, the validity of the lipidomics results (specimen quality, analytical methods) may require debate and further investigation.

Major points

1. A major concern in lipidomics has been the reproducibility of results across trials and institutions (PMID: 35780914, 29959947). One factor contributing to this issue is specimen quality. Blood samples, in particular, can exhibit significant fluctuations in lipid levels depending on how they are handled from post-collection to lipid extraction. Given the limited number of lipidomics studies using menstrual blood, meticulous handling of these specimens is crucial.

1-1. Do the authors have insights on how blood lipid levels might change during collection, drying, and storage at -30°C? Is the protocol designed to minimize these variations? Additionally, the time and method from collection to drying should be described in greater detail in the manuscript.

1-2. Phospholipid oxidation is significantly enhanced if specimens are allowed to stand after drying. Can the authors discuss the implications of analyzing oxidized lipids in dried blood samples?

2. The authors analyzed the MS data using MzMine 2.3 and LipidMatch, normalizing the peak areas of the target lipids by sum of the peak areas. However, how reliable is this quantification method? For CL 16:0_18:0_22:5_22:6 and PE P-16:0/18:1, the best candidates identified by the authors, it is recommended to validate the results by target analysis with appropriate internal standards and multiple reaction monitoring (MRM).

3. It is interesting to note that cardiolipin was elevated in the patient's menstrual blood, suggesting a link to mitochondrial function. However, the significant increases were observed in minor CL molecular species, such as CL 16:0_18:0_22:5_22:6, rather than major molecular species like CL 18:2_18:2_18:2_18:2. Can the authors provide a discussion on this observation?

4. Samples derived from patients with endometriosis at different stages were analyzed in this study. Could the levels of candidate lipids be correlated with disease stages or symptoms?

Minor points

5. Cardiolipin was detected in this study as [CL-2H], although the [CL-H] form typically ionizes more efficiently. Is there a specific reason why [CL-H] was not detected in your analysis?

6. The notation X (Y;Z) appears frequently. It seems that X represents the mean value, while Y and Z denote the minimum and maximum values, respectively. To enhance clarity, it might be better to change this notation.

Author Response

The authors have found several lipid molecular species that may define endometriosis by lipidomics analysis of menstrual blood samples. These findings would be novel and important, because there have been many reports on lipidomics using plasma/serum from patients with endometriosis, but not menstrual blood. However, the validity of the lipidomics results (specimen quality, analytical methods) may require debate and further investigation.

Major points

2.1 A major concern in lipidomics has been the reproducibility of results across trials and institutions (PMID: 35780914, 29959947). One factor contributing to this issue is specimen quality. Blood samples, in particular, can exhibit significant fluctuations in lipid levels depending on how they are handled from post-collection to lipid extraction. Given the limited number of lipidomics studies using menstrual blood, meticulous handling of these specimens is crucial.

2.1-1. Do the authors have insights on how blood lipid levels might change during collection, drying, and storage at -30°C? Is the protocol designed to minimize these variations? Additionally, the time and method from collection to drying should be described in greater detail in the manuscript.

Answer. Thank you for your in-depth study of our work. Our protocol was designed in accordance to He D., et al., 2023 and Prentice P., eat al., 2013 studies of DBS metabolites (including lipids) stability at -20 oC - -80 oC storage (references added). Additional information about the samples processing before storage was included: “Blood drops were air-dried for two hours in the dark at room temperature before being placed in an envelope for transportation.”. To correct the analytical variance, the QC and blank samples were used in accordance to guidelines provided by mQACC (https://www.mqacc.org/): “In accordance to guidelines provided by mQACC (https://www.mqacc.org/) blank and quality control samples were included in batch. The quality control samples were prepared by pooling of 10 µl from each study sample.” and “In positive ion mode, almost half of lipids have between injection variation less than 10%. In negative ion mode, more than a half of lipids exhibited between injection variation less than 10%.”..

2.1-2. Phospholipid oxidation is significantly enhanced if specimens are allowed to stand after drying. Can the authors discuss the implications of analyzing oxidized lipids in dried blood samples?

Answer: Thank you for your suggestion. We added the following information in discussion section: “However, increased levels of oxidized phospholipids may be an artifact of sample collection, namely the long stand after drying. In this work, the drying process was strictly controlled, which allows minimizing this potential confounder.”.

2.2. The authors analyzed the MS data using MzMine 2.3 and LipidMatch, normalizing the peak areas of the target lipids by sum of the peak areas. However, how reliable is this quantification method? For CL 16:0_18:0_22:5_22:6 and PE P-16:0/18:1, the best candidates identified by the authors, it is recommended to validate the results by target analysis with appropriate internal standards and multiple reaction monitoring (MRM).

Answer: The method applied, MzMine 2.3 and LipidMatch, is reliable and reproducible for a pilot study. For example, QC-deviation of CL 16:0_18:0_22:5_22:6 was 11% in case of non-normalized values and 3% in case of normalized values and QC-deviation of PE P-16:0/18:1 was 8% in case of non-normalized values and 3% in case of normalized values (data included in the article). Moreover, the amount of samples was not enough to perform additional experiments (target HPLC-MRM-MS analysis with internal standards). In future, we plan to validate the markers proposed with a new set of samples.

2.3. It is interesting to note that cardiolipin was elevated in the patient's menstrual blood, suggesting a link to mitochondrial function. However, the significant increases were observed in minor CL molecular species, such as CL 16:0_18:0_22:5_22:6, rather than major molecular species like CL 18:2_18:2_18:2_18:2. Can the authors provide a discussion on this observation?

Answer: Thank you for your remark. The following text was added to Discussion section: “We can only guess why the detected pronounced changes in cardiolipin levels affect minor lipids rather than major molecular species like CL 18:2_18:2_18:2_18:2. Perhaps the reason lies in the disruption in endometriosis of the fine mechanisms regulating the composition of the inner mitochondrial membrane of epithelial cells, shedding during menstruation..”

2.4. Samples derived from patients with endometriosis at different stages were analyzed in this study. Could the levels of candidate lipids be correlated with disease stages or symptoms?

Answer: We added the information about lipid correlation with stages in the article (Figure 1B). Moreover, CL 16:0_18:0_22:5_22:6 positively correlated with endometriosis stage (R=0.56, p<0.001).

Minor points

2.5. Cardiolipin was detected in this study as [CL-2H], although the [CL-H] form typically ionizes more efficiently. Is there a specific reason why [CL-H] was not detected in your analysis?

Answer: Cardiolipin in our analysis (ESI, negative mode) ionized more effectively with the loss of two protons. It is not a unique case. In particular, Garett et al. presented more abundance of [CL -2H+]-2 than [CL – H+]-  (doi.org/10.1016/S0076-6879(07)33012-7). Moreover, Xianlin Han supposed, than in the most cases twice charged cardiolipin ion is predominant (doi.org/10.1002/9781119085263).

2.6. The notation X (Y;Z) appears frequently. It seems that X represents the mean value, while Y and Z denote the minimum and maximum values, respectively. To enhance clarity, it might be better to change this notation.

Answer: We added description of notation X (Y;Z) in Materials and methods section: “Relative levels of lipids are presented as Me(Q1;Q3), where Me is median, Q1 is the first quartile, Q3 is the third quartile.” and in the caption of Table 1: “Values are presented as Me(Q1;Q3), where Me is median, Q1 is the first quartile, Q3 is the third quartile.”.

Reviewer 3 Report

Comments and Suggestions for Authors

Even though it is a pilot study, it is a gold standard study for early diagnosis and/or more effective prognosis in endometriosis. We know that women with endometriosis suffer a lot, not just physically, but psychologically. This is due, in most cases, to the discovery that the disease is already related to infertility. Therefore, continued exploration of biomarkers present in menstrual blood may actually and potentially usher in improved diagnostic approaches and more effective treatment tactics in endometriosis. Studies involving the lipid relationship in endometriosis conditions are increasing, mainly with miRs because some miRs with increased expression in endometriosis are related to decreased apoptosis of granulosa cells. Thus, lipid involvement is crucial in the development of endometriosis and, therefore, in its early diagnosis and/or treatment leading to a reduction in infertility caused by endometriosis.

Yours sincerely,

Author Response

Even though it is a pilot study, it is a gold standard study for early diagnosis and/or more effective prognosis in endometriosis. We know that women with endometriosis suffer a lot, not just physically, but psychologically. This is due, in most cases, to the discovery that the disease is already related to infertility. Therefore, continued exploration of biomarkers present in menstrual blood may actually and potentially usher in improved diagnostic approaches and more effective treatment tactics in endometriosis. Studies involving the lipid relationship in endometriosis conditions are increasing, mainly with miRs because some miRs with increased expression in endometriosis are related to decreased apoptosis of granulosa cells. Thus, lipid involvement is crucial in the development of endometriosis and, therefore, in its early diagnosis and/or treatment leading to a reduction in infertility caused by endometriosis.

Answer. Thank you very much for your kind evaluation of our research.

Round 2

Reviewer 2 Report

Comments and Suggestions for Authors

The authors properly revisited the manuscript.

Author Response

Comments and Suggestions for Authors:

The authors properly revisited the manuscript

Answer: Thank you very much for reviewing our manuscript.